# Factors Determining the Agreement between Aerobic Threshold and Point of Maximal Fat Oxidation: Follow-Up on a Systematic Review and Meta-Analysis on Association

**DOI:** 10.3390/ijerph20010453

**Published:** 2022-12-27

**Authors:** Carlo Ferri Marini, Philippe Tadger, Isaac Armando Chávez-Guevara, Elizabeth Tipton, Marco Meucci, Zoran Nikolovski, Francisco Jose Amaro-Gahete, Ratko Peric

**Affiliations:** 1Department of Biomolecular Sciences, Division of Exercise and Health Sciences, University of Urbino Carlo Bo, 61029 Urbino, Italy; 2Real World Evidence, IQVIA, 3600 Genk, Belgium; 3Department of Chemical Sciences, Biomedical Sciences Institute, Ciudad Juarez Autonomous University, Chihuahua 32310, Mexico; 4Department of Statistics and Data Science, Northwestern University, Evanston, IL 60208, USA; 5Department of Health and Exercise Science, Appalachian State University, Boone, NC 28608, USA; 6Faculty of Kinesiology, University of Split, 21000 Split, Croatia; 7Department of Physiology, Faculty of Medicine, University of Granada, 18001 Granada, Spain; 8PROmoting FITness and Health through Physical Activity Research Group (PROFITH), Department of Physical and Sports Education, School of Sports Science, University of Granada, 18011 Granada, Spain; 9EFFECTS-262 Research Group, Department of Physiology, School of Medicine, University of Granada, 18016 Granada, Spain; 10Centro de Investigación Biomédica en Red Fisiopatología de la Obesidad y Nutrición, Instituto de Salud Carlos III, 28029 Madrid, Spain; 11Instituto de Investigación Biosanitaria de Granada, 18014 Granada, Spain; 12Department for Exercise Physiology, Orthopedic Clinic Orthosport, 78000 Banja Luka, Bosnia and Herzegovina

**Keywords:** peak fat oxidation, aerobic threshold, exercise, agreement

## Abstract

Regular exercise at the intensity matching maximal fat oxidation (FAT_max_) has been proposed as a key element in both athletes and clinical populations when aiming to enhance the body’s ability to oxidize fat. In order to allow a more standardized and tailored training approach, the connection between FAT_max_ and the individual aerobic thresholds (AerT) has been examined. Although recent findings strongly suggest that a relationship exists between these two intensities, correlation alone is not sufficient to confirm that the intensities necessarily coincide and that the error between the two measures is small. Thus, this systematic review and meta-analysis aim to examine the agreement levels between the exercise intensities matching FAT_max_ and AerT by pooling limits of agreement in a function of three parameters: (i) the average difference, (ii) the average within-study variation, and (iii) the variation in bias across studies, and to examine the influence of clinical and methodological inter- and intra-study differences on agreement levels. This study was registered with PROSPERO (CRD42021239351) and ClinicalTrials (NCT03789045). PubMed and Google Scholar were searched for studies examining FAT_max_ and AerT connection. Overall, 12 studies with forty-five effect sizes and a total of 774 subjects fulfilled the inclusion criteria. The ROBIS tool for risk of bias assessment was used to determine the quality of included studies. In conclusion, the overall 95% limits of agreement of the differences between FAT_max_ and AerT exercise intensities were larger than the a priori determined acceptable agreement due to the large variance caused by clinical and methodological differences among the studies. Therefore, we recommend that future studies follow a strict standardization of data collection and analysis of FAT_max_- and AerT-related outcomes.

## 1. Introduction

Lipids and carbohydrates are the main substrates utilized during exercise with their absolute and relative contribution being influenced by sex, diet, exercise intensity and duration, fitness level, and time of day [1]. During exercise, the fat oxidation rate follows a curvilinear pattern, and as the exercise workload progressively augments, fat oxidation increases to its maximal oxidation rate called FAT_max_, a term used to describe an intensity at which the energy contribution from lipids is at its highest [2,3]. As the workload progresses toward heavy-to-severe exercise intensities, lipid oxidation decreases and carbohydrates become the predominant energy substrate with lipids becoming negligible [4]. With the body’s limited storage capacity for carbohydrates, optimizing substrates’ availability is recognized as one of key factors limiting the performance during high intensity and prolonged exercise [5]. 

Regular exercise at FAT_max_ is of high interest for athletes as training at this intensity can improve lipid oxidation during exercise, which is associated with the performance of endurance-trained individuals [5,6]. Moreover, with the current obesity epidemic representing a serious global medical issue, exercising at FAT_max_ has also gained attention among public health professionals and has been recommended for treating several chronic metabolic health issues [5,6]. In patients with chronic metabolic diseases, FAT_max_ training provides a beneficial alternative for optimizing fat oxidation during exercise, which has proven to be an effective strategy for improving body composition, insulin sensitivity, and oxidative capacity in patients with obesity, metabolic syndrome, and type 2 diabetes mellitus [7,8]. Concretely, (i) the progressive activation of type II muscle fibers that exhibit a high glycolytic activity, (ii) the rise in circulating levels of glucagon and catecholamines that promote glycogen partitioning in the liver and skeletal muscle, (iii) the reduction in carnitine levels at the sarcoplasm, and (iv) the inhibition of the enzyme carnitine palmitoyltransferase I (CTP1) due to pH modifications are all considered physiological and molecular mechanisms explaining the effects of FAT_max_-based training [9,10,11]. 

The accurate prescription of individualized exercise intensities is a complex and still controversial task [12,13,14]. The traditional approach used to prescribe and monitor exercise intensity is based on the use of physiological parameters, such as oxygen uptake (VO_2_) or heart rate (HR), expressed either in absolute (e.g., mL/kg/min and beats per minute (b/min), respectively) or relative (e.g., percentages of their maximal values, %VO_2max_ and %HR_max_, respectively) values [13]. Although these methods are commonly used and widely accepted by the scientific community, they have been shown to produce a high inter-individual variability in exercise intensity [15]. Indeed, the high inter-individual variability in the metabolic responses resulting from exercise prescription methods based on fixed percentages of maximum values yielding FAT_max_ (35–75%VO_2max_ and 45–65%HR_max_) has been revealed [1,8,16]. Previous studies have postulated that, when exercise intensity is expressed as a fixed percentage of maximal values, it may not accurately reflect the responses of the human body [17,18,19]. Therefore, some authors recommend using exercise prescription based on the workload corresponding to individualized metabolic thresholds, allowing for more accurate intensity standardization [13,14,18]. In this way, specific transition points between metabolic pathways are defined, allowing for a prescription of individualized exercise while overcoming differences in an individual’s phenotype [14].

Two metabolic thresholds have been traditionally identified, defining three distinct zones of energy production [20]. In this paper, to avoid any confusion, the terminology provided by Meyer et al. [20] is preferred, with the term aerobic threshold (AerT) used for the first threshold. The modification of the muscle fibers’ pH induced by the rise in glucose oxidation and lactic acid accumulation activates the Acid-sensing ion channels (ASICs), increasing afferent neuron activity, which signals the cardiovascular and respiratory centers to elevate the blood pressure, heart rate, left ventricular contractility, and ventilation [21,22]. This physiological acute adaptation of humans is defined as the “metaboreflex”, and evidences that metabolic transition points could be assessed by the measurement of cardiorespiratory parameters (e.g., heart rate and pulmonary ventilation), with a detailed description of appropriate methodological approaches used for their identification provided elsewhere [23].

In order to increase the accuracy and effectiveness of training prescription [13,16], scientists have investigated the relationship between FAT_max_ and AerT in different populations [24,25,26]. However, over the last few decades, the scientific literature in this field has produced controversial results, leaving this topic open for debate. Overall, large variations in relationship strength have been revealed, varying from almost no relationship to an almost absolute relationship in different populations [16,24,25,26]. A recent systematic review tried to provide some conclusions and eventually confirmed the existence of a strong association between the exercise intensities matching FAT_max_ and AerT [27]. It is worth noting that the authors highlighted that the association between FAT_max_ and AerT seems to be influenced by the variations in the studies’ methodological approaches, fore mostly used for the estimation of both physiological biomarkers [27]. Despite the conclusions drawn by Peric et al. [27], the results on the correlations and standardized mean difference failed to investigate whether FAT_max_ and AerT intensities can be used interchangeably and yield an acceptably low error at the individual level. Even though the presence of a strong association between FAT_max_ and AerT confirms that both outcomes are correlated, knowing the error for a given individual—which is of paramount importance for aerobic exercise prescription purposes—remains unknown. For example, Rynders et al. [28] reported a large error with wide limits of agreement (LoAs) between intensities matching FAT_max_ and AerT despite observing a strong association between both parameters. On the contrary, Gonzales-Haro [29] revealed a weak association followed by wide LoAs. Therefore, although the exercise intensities that elicit FAT_max_ and AerT have been thoroughly investigated [1,30,31], the agreement and interchangeability of FAT_max_ and AerT is still a controversial topic that requires further research. 

Hence, the purpose of this systematic review and meta-analysis is to examine the agreement between the exercise intensities at FAT_max_ and AerT and identify relevant moderators affecting their agreement.

## 2. Methodology

### 2.1. Study Design

The present systematic review was registered in both PROSPERO (International Prospective Register of Systematic Reviews; CRD42021239351) and the ClinicalTrials.gov website (NCT03789045). We used the Preferred Reporting Items for Systematic Reviews and Meta-Analyses (PRISMA) checklist/flow-diagram in order to (i) structure this document, (ii) effectively explain the methodology, and (iii) systematically report the search results [32].

### 2.2. Search Strategy

A systematic search in MEDLINE (via PubMed) and Google Scholar was conducted for the identification of studies investigating exercise intensities at FAT_max_ and AerT in humans. For this purpose, we used a Boolean Logic limiting the search results through different operators (i.e., AND/OR) in order to identify the manuscripts that reported the above-mentioned biomarkers. The following terms were combined to account for differences in the terminology [31]: (“fat oxidation” OR “maximal fat oxidation” OR “optimal fat oxidation” OR “peak fat oxidation”) AND (“aerobic threshold” OR “anaerobic threshold” OR “ventilatory threshold” OR “lactate threshold” OR “metabolic threshold” OR “gas exchange threshold”). Two reviewers (PR and NZ) independently developed and conducted the search in accordance with the PRISMA statement [32]. Moreover, given that the FAT_max_ concept was initially referred to in 2001 [4], the search included publications from 1 January 2001 to 30 March 2022.

### 2.3. Inclusion/Exclusion Criteria and Risk of Bias Assessment

In the present systematic review, only original studies (i.e., randomized and non-randomized controlled trials, cohort studies, case–control studies, and cross-sectional studies) in the form of full text, abstracts, or congress presentations were included. Selected articles were classified by using the defined population, intervention, comparison, outcomes, and study design (PICOS) [33]. Moreover, the studies included had to report (i) AerT and FAT_max_ intensities values (i.e., mean ± SD of difference) or (ii) the levels of agreement of Bland–Altman analysis. If one of the requirements failed to be reported or could not be determined from the full-text, the corresponding author was contacted and asked to provide either the missing information or the individual subjects’ raw data, allowing us to compute the data. Nevertheless, independent of its form, all studies required a clear and reproducible description of the methods used to determine both AerT and FAT_max_ to be considered eligible. Case reports, editorials, reviews, and opinion papers were excluded. In addition, only studies that included participants with no evidence of any metabolic, pulmonary, or cardiovascular diseases (i.e., conditions potentially affecting substrate utilization) were considered acceptable. No other restrictions to participants’ characteristics were introduced other than age (18–60 years). Report criteria required studies to be written in English and to be published in a peer-reviewed journal. The grading of recommendations, assessment, development, and evaluation (GRADE) approach was applied to define the quality of a body of evidence for selected studies [34]. A tool for assessing the risk of bias in systematic reviews, ROBIS, was used for critical evaluation of the collected data [35].

### 2.4. Data Extraction

Data extraction of the articles that met the inclusion criteria was performed in an unblinded standardized way, by four independent reviewers (PR, TP, IACG, and FMC), using a data extraction Excel spreadsheet (Microsoft, Redmond, WA, USA). Any disagreements about data extraction were resolved by consensus between the reviewers.

### 2.5. Statistical Analysis

All statistical analyses were performed with R software (version 4.0.4) (The R Foundation, Vienna, Austria) by using metafor (version 3.8.1), clubSandwich (version 0.5.8), forester (version 0.2.0), and outliers (version 0.15) packages [36,37,38,39]. The analyses of agreement between the AerT and FAT_max_ were assessed by using a Bland–Altman meta-analytic approach [40]. Due to the differences between the scales of the measurement units used to identify FAT_max_ and AerT, the agreement analyses were performed separately for each of the measurement units identified. The primary outcome of the Bland–Altman meta-analyses was the limits of agreements (LoAs) of the differences in exercise intensities matching AerT and FAT_max_, and secondarily their mean bias. For each study, the mean bias (i.e., the mean difference between AerT and FAT_max_) and the standard deviation (SD) of the bias were retrieved or imputed from the AerT and FAT_max_ means, SDs, and Pearson correlation (*r*) or bias LoA of the selected studies. In one study [41], due to the lack of information needed to estimate the SD of the bias, the *r* between AerT and FAT_max_ was estimated as the median correlation of the studies having the same measurement unit (i.e., *r* = 0.76) [27], which was used to compute the SD of the bias. Then, the mean and SD of the biases were used to calculate the Bland–Altman meta-analyses LoAs.

Specifically, both the average within-study variation (i.e., the averages across the studies of the mean biases (δ) and of the within-study variations in the differences (σ^2^)) and the between-study variation (i.e., variation in the mean bias across studies (τ^2^)) were used to account for different sources of variability and to compute population LoAs as follows: δ ± 2√(σ^2^ + τ^2^) [42,43]. Additionally, the outer 95% confidence intervals (CIs) for pooled limits of agreement were computed to account for the LoA uncertainty. 

Estimating the LoAs and their 95% CIs required several steps. First, the study mean biases were modeled using a (multilevel) random-effects model, providing estimates of δ and τ^2^. Second, the study (logged) variances were modeled using separate (multilevel) random effects, providing an estimate of σ^2^. The standard errors of estimates of δ and σ^2^ were estimated using Robust Variance Estimation (RVE), while the standard error of τ^2^ was estimated using a formula previously provided [40]. Third, these estimates and their standard errors were combined in order to estimate the LoAs and their 95% CIs using estimators proposed elsewhere [40].

In analyses in which there was only a single effect size from each sample, standard random-effects models were used [42]. In analyses where multiple effect sizes originated from the same sample, a correlated hierarchical effects model was used [43]. This required imputing the correlation (RHO) between these effect sizes, as it was unreported; hence, we set RHO to be the median correlation between AerT and FAT_max_ (e.g., median *r* = 0.77 for mL/kg/min and 0.76 for %VO_2max_, respectively) [27]. In order to guard against possible misspecification, we used RVE to estimate the standard errors [44,45]. Additionally, sensitivity analyses with RHO equal to 0.1 were also performed [46]. The same analyses were performed for several subgroups, grouped according to methodological and clinical difference in order to account for sources of potential heterogeneity [47]. When a subgroup presented only one effect size, the LoAs were estimated as mean bias ± 1.96 SD of the bias [40].

The Grubbs test was used to evaluate the presence and effect of the outliers in the current data, and a cut-point for such outliers’ recognition was estimated [48]; such outliers were excluded to recalculate an overall pool estimation, as a sensitive analysis.

Finally, a priori determination of a clinical and physiologically practical acceptable range of the LoA was based on the concept of FAT_max_ zone (i.e., range of exercise intensities with fat oxidation rates within 10% of fat oxidation rates at FAT_max_) [4], which was estimated using a meta-analytical approach (Appendix A) [49]. If the LoA failed within the FAT_max_ zone reported for the corresponding measurement unit, then AerT could be used for optimizing fat oxidation with an acceptable error. The R software code used in this study is freely available in Appendix A.

## 3. Results

### 3.1. Descriptive Results

The systematic search retrieved a total of 72,205 papers with 18,420 identified as duplicates. After the screening of the paper’s title and language, a total of 675 manuscripts were retained. Finally, articles’ abstracts were screened and a total of 89 records were selected to be read in full as potentially eligible. An additional 77 articles were excluded after reading the full text as they did not meet the inclusion criteria. The literature search and study selection process are reported using the PRISMA statement in Figure 1. The internal validity of included studies, as obtained by the ROBIS tool, showed no risk of bias within the included data. Overall, twelve papers and a total of 45 effect sizes were included in the present study. Due to anticipated heterogeneity, four investigators (PR, TP, TE, and FMC) independently performed the identification of potentially relevant moderators and their subgroups, allowing sources of variation to be investigated. The identified moderator variables, categorized by differences in characteristics of the studies (i.e., methodological diversity) and by study populations (i.e., clinical diversity), are reported in Table 1. 

### 3.2. Clinical Differences

When the selected studies were examined for their clinical diversity, two moderators were identified: (i) sex and (ii) physical activity level, each having two different subgroups (Table 1). The included papers evaluated 774 participants, out of which 299 (38.6%) were females and 545 (61.4%) were males. Within the twelve included studies, seven studies (58.3%) used males as primary participants, whereas one (8.3%) study tested only females. In four (33.3%) studies, both sexes were evaluated. When classified by physical activity level, seven studies (58.3%) examined active subjects (e.g., high-level cyclists, moderately trained cyclists, triathlon athletes, endurance runners, sprinters, and ball game athletes), whereas five (41.7%) studies examined inactive participants (e.g., sedentary and obese). Out of the overall included participants, 311 (40.2%) subjects were active while 463 (59.8%) were inactive.

### 3.3. Methodological Differences

When the selected studies were examined for their methodological diversity, six moderators with relevant subgroups were identified (Table 1). The primary moderator identified was related to the characteristic of measurement unit used to assess the exercise intensities matching FAT_max_ and AerT and, consequently, agreement. Due to the high variability present, only this moderator is presented according to the effect sizes (Table 1). In three (6.7%) effect sizes, percentage of maximal HR (%HR_max_) was used, whereas in five (11.1%) cases, HR beats per minute (b/min) was used. For two (4.4%) effect sizes, absolute VO_2max_ values were used (L/min). A %VO_2max_ was used in 12 (26.7%) cases, whereas relative VO_2max_ values (mL/kg/min) were used in 13 cases (28.9%). As for the ergometry type used, cycle ergometry was the preferred method in seven (58.3%) studies, whereas a treadmill was employed in five (41.7%). When considering the methods used to determine the AerT, four (33.3%) studies preferred the lactate method, whereas gas analysis was used in six (50%). On two (16.7%) occasions, the studies used both methods. Regarding the protocols used to assess VO_2max_/AerT, our results showed that all studies favored a graded exercise test (GXT) type protocol. Even though the included studies used GXT protocols ranging anywhere from 1 min to 5 min, no two studies implemented the same GXT protocol. Hence, due to high variability and to assure appropriate statistical analysis, stage lengths ≤3 min were classified as short, whereas those >3 min were considered as long. Ten (83.3%) studies preferred shorter stages while longer stages were used in only one study (8.3%). One (8.3%) study used both short and long stages. FAT_max_ identification was performed by using visual inspection of the appropriate plots in seven (58.3%) studies, while five (41.7%) studies identified this occurrence by using a mathematical model. Lastly, ten (83.3%) studies determined FAT_max_ during the assessment of VO_2max_/AerT, whereas two (16.7%) studies used an additional test.

### 3.4. Agreement Analyses

Assuming the normal distribution of differences between AerT and FAT_max_, expressed as mL/kg/min and %VO_2max_, the effect of pooling study distributions can be seen in Figure 2. For this graphical representation, each individual study’s distribution is given in grey, with the overall pooled distribution given in black. A benefit of this visualization is that it makes it clear that the focus of the analysis is not simply on the average bias (delta) but rather on LoA. The results (LoAs and mean biases) of all studies (with or without potential outliers) and subgroup Bland–Altman meta-analyses for the analyzed measurement units are reported in Figure 3. Additional agreement indicators are presented in Appendix A.

### 3.5. Sensitivity Analysis

Due to results of the sensitivity analyses, the overall estimates were also performed without two effect sizes for mL/kg/min and one for %VO_2max_, respectively (Table 1 and Figure 3). Additionally, in the follow-up sensitivity analysis, the results of all the above-mentioned analyses were computed with an estimated correlation between repeated measures deriving from the same individual of RHO = 0.1 (Appendix A).

## 4. Discussion

In the present study, the agreement between FAT_max_ and AerT was investigated primarily by looking at the LoAs, which provides information about the expected range including 95% of the differences in exercise intensity at FAT_max_ and AerT, and secondarily on the bias. The main result of this study is that, regardless of the measurement unit used, the overall LoAs were larger than the a priori clinically acceptable cut-off. Therefore, the observed results imply that researchers and practitioners might experience a clinically relevant error (i.e., difference between the two intensities) at the individual level when aiming to interchangeably use FAT_max_ and AerT exercise intensities. 

Follow-up subgroup analyses show that, when VO_2_ is expressed in mL/kg/min, the main factors (moderators) affecting the agreement between FAT_max_ and AerT are the physical activity level (e.g., active vs. inactive), the type of ergometer used (e.g., treadmill vs. cycle), the analytical process used to identify the AerT (e.g., lactate vs. gas analysis), and the duration of GXT exercise stages (e.g., short vs. long). However, the same trends were not observed in the agreement analyses between FAT_max_ and AerT when exercise intensities are expressed in %VO_2max_ where apparently none of the moderators have an effect on the agreement. 

The results obtained for the physical activity level (using mL/kg/min) moderator could be due to the fact that the subgroup including inactive individuals was characterized by studies that included participants with a similar lower fitness level (i.e., sedentary and obese), creating a narrower range of possible exercise intensities (resulting in narrower LoAs) for this subgroup. Contrarily, the fitter individuals were characterized by a wider range of exercise intensities due to a higher VO_2max_, and a higher heterogeneity due to different individual characteristics caused by the practice of different sports types (i.e., endurance athletes, sprinters, and team sports). 

On the other hand, the results observed for the physical activity level (using %VO_2max_) moderator could be related to the exercise intensity range and the fact that it can always reach 100% of %VO_2max_, resulting in a greater normalization of the exercise intensity range, independently from the physical activity type. Therefore, the narrower LoAs observed for the exercise intensity expressed in mL/kg/min might be related to a lack of standardization in this particular method and its smaller scale caused by participants with lower fitness level. This could be further explained by both physiological and methodological factors. From a physiological standpoint, the fitness level and the training exercise intensity can dictate the levels of improvement of the AerT [56]. Specifically, in endurance-trained athletes, the AerT occurs at higher relative exercise intensities, compared to other athletes [20]. Therefore, the discrepancies in the training regimens (e.g., cycling vs. running), and the fitness level (e.g., moderately trained vs. athletes) of active individuals considered in this meta-analysis could lead toward a larger inter-subject variability in AerT, due to variations in sport-specific physiological demands. Indeed, the intra-individual coefficient of variation (CV) of the AerT was higher in the studies performed on active individuals (CV: 3–54%) [16,29,41,51,54] in comparison to those studies performed on inactive subjects (CV: 12–45%) [25,28,50,53,55]. Furthermore, similar factors have been reported to affect the FAT_max_ [5,30]. Therefore, considering that FAT_max_ and AerT tend to be affected by similar factors (e.g., individuals with higher fitness level tend to have higher FAT_max_ and AerT) and, as shown in a recent systematic review [27], are correlated with each other, the existence of an association between AerT and FAT_max_ should cause a shift in both parameters simultaneously when certain factors change. Consequently, the factors that affect the exercise intensity at which FAT_max_ and AerT appear do not necessarily affect their agreement. According to our data, such a reaction was only observed in inactive subjects, whereas active individuals failed to demonstrate such a connection, leaving this open for further debate.

As for the type of ergometer used, we believe a similar explanation could be the potential answer. Indeed, the studies that preferred the treadmill in the current systematic review tended to have subjects with a lower fitness level (Table 1), hence providing the basis for a narrower LoA in this subgroup. 

A broad LoA of the differences between FAT_max_ and AerT in the long GXT stages subgroup, independently from the measurement unit used, could be due to the presence of only two ESs deriving from the same study when VO_2_ was expressed as mL/kg/min [50] and when VO_2_ was expressed as %VO_2max_ [29], yielding a less robust representation of the LoAs. Hence, future studies with variations in the GXT stages’ length are needed to allow a more detailed insight.

As for the analytical process used to assess AerT, three of the studies that evaluated active individuals determined the AerT by the measurement of blood lactate levels [29,41,50], while three studies determined the AerT by gas analysis [16,51,54]. Such inconsistency in the analytical procedures may partly explain the larger heterogeneity observed in active individuals and provide a basis for the narrower LoAs observed. In fact, research studies that used blood lactate concentration to identify the AerT reported a lower agreement between the VO_2_ at FAT_max_ and at AerT compared to those using gas analysis (Figure 3a). It is worth noting that all the studies that evaluated the AerT by gas analysis used the universal method: the ventilatory equivalents of oxygen approach [23], whereas different analytical procedures were used for the definition of the AerT by lactate. Moreover, the studies of Bircher et al. [24] and Michallet et al. [26] defined the AerT as the exercise intensity at which blood lactate increased 0.5 mmol above baseline. This analytical procedure previously showed a poor agreement and association with the AerT on trained individuals [57] and men with obesity [55]. Indeed, studies defined the AerT as the exercise intensity at which lactate started to accumulate, showing a sustained increment in the bloodstream [28,29,55]. This analytical procedure showed a stronger association and a good agreement between AerT and FAT_max_ in both trained and sedentary individuals [57,58]. Thus, considering that the manipulations of the lactate method can affect the validity of the lactate threshold [59], future studies need to corroborate the influence of the analytical procedure used to define the lactate threshold when examining the connection between FAT_max_ and AerT. 

It is noticeable that when exercise intensity is expressed in mL/kg/min, three of the five studies evaluating inactive subjects examined patients with obesity [24,50,55], suggesting that, in this population, using AerT to prescribe exercise can yield intensities relatively within the FAT_max_ zone, with small differences between FAT_max_ and the AerT intensities. Additional studies investigating the agreement between the lactate threshold and FAT_max_ in inactive individuals with obesity when %VO_2max_ is used are needed as only Chávez-Guevara et al. [55] defined the AerT by using both lactate and gas analysis. The analysis of blood lactate concentration may be a valuable alternative to gas exchange as it allows clinicians and healthcare practitioners to identify FAT_max_ in field-based settings. From a physiological point of view, the connection between the FAT_max_ and AerT can be further explained by the metaboreflex paradigm. This model explains that both fatty acid oxidation and ventilatory response are regulated by the metabolic transitions induced by muscle acidosis, which also results in lactate accumulation in the muscle [10,22]. The physiological background provided by this paradigm supports the observed association and agreement between the FAT_max_ and AerT. This is a very important finding because chronic exercise training at FAT_max_ has been shown to improve physical fitness, metabolic flexibility, and cardiovascular function in patients with obesity [6,8].

There are some limitations that need to be considered for a proper interpretation of the findings reported in this meta-analysis. In the first place, it was not possible to finely investigate every methodological study difference. Indeed, in order to analyze the methodological within- and between-studies differences and their influence on the agreement, identified moderators had to be divided into subgroups. Even though our results revealed a direct influence of certain moderators (i.e., physical activity level, the type of ergometer, the analytical process used to identify the AerT, and the duration of GXT exercise stages) on the agreement, a limited number of the included studies and a large presence of methodological between-studies differences failed to allow a more detailed subgrouping, leaving some questions unanswered (i.e., variations within the short GXT stages, variation within inactive subjects, etc.). Further on, only those studies performed in young and middle-aged adults were included into this meta-analysis. Previously, Tolfrey et al. [59] reported a good agreement between the FAT_max_ and AerT defined by lactate analysis in untrained adolescents. Moreover, Meucci et al. [60] reported a strong relationship between FAT_max_ and AerT defined by gas analysis in adolescents. Both studies used %HR_max_ to identify exercise intensity matching FAT_max_ and AerT. To the best of our knowledge, no studies have investigated the agreement or association between the FAT_max_ and AerT in elders, warranting further research. In the second place, studies have reported a substantial day-to-day variation in the FAT_max_ (CV: 3–26%) [61,62]; hence, further studies need to investigate the reliability of the agreement between FAT_max_ and AerT. As previously mentioned, our results showed narrower LoAs for mL/kg/min when compared to %VO_2max_. It is worth noting that comparing LoAs coming from different measurement units with different scales will always lead toward a higher range if % is used. Therefore, between-measurement-unit differences should be taken with care and not directly compared. Indeed, when interpreting the main factors (subgroups) moderating the agreement (e.g., physically inactive vs. active), a probable cause of our observation could be due to the smaller scale. Therefore, the lack of a visible important effect of a subgrouping variable when a different measurement unit is used is presumably due to the inability of one of the measurement units to standardize the intensity in the different subgroups, which could be due to the different scales of the units and not directly due to the moderator (subgroup) effect. Another limitation of the obtained results is that the agreement for all measurement units used to prescribe aerobic exercise intensity was not assessed. As in the case of the moderators, this was due to the limited number of studies presenting agreement for certain measurement units (e.g., bpm, %HR_max_, and L/min; see Table 1). Therefore, future studies assessing the agreement between FAT_max_ and AerT using alternative parameters to those analyzed in the present study are recommended. Lastly, a priori identification of the clinically relevant FAT_max_ zone was partially based on the studies included in this study, but mostly originated from different studies where the needed data for such calculation were available. Hence, we applied the FAT_max_ zone obtained in different groups of subjects, which might have resulted in a potential bias of our results.

## 5. Conclusions

Our results showed that the agreement between FAT_max_ and AerT intensities is low and seems to be primarily affected by the measurement unit used to define exercise intensity matching AerT and FATmax followed by the influence of certain moderators. Therefore, prescribing exercise based on the AerT to maximize fat oxidation should be performed cautiously because, due to the high variability, there is a substantial possibility that AerT could fall well outside the FAT_max_ zone.

## Figures and Tables

**Figure 1 ijerph-20-00453-f001:**
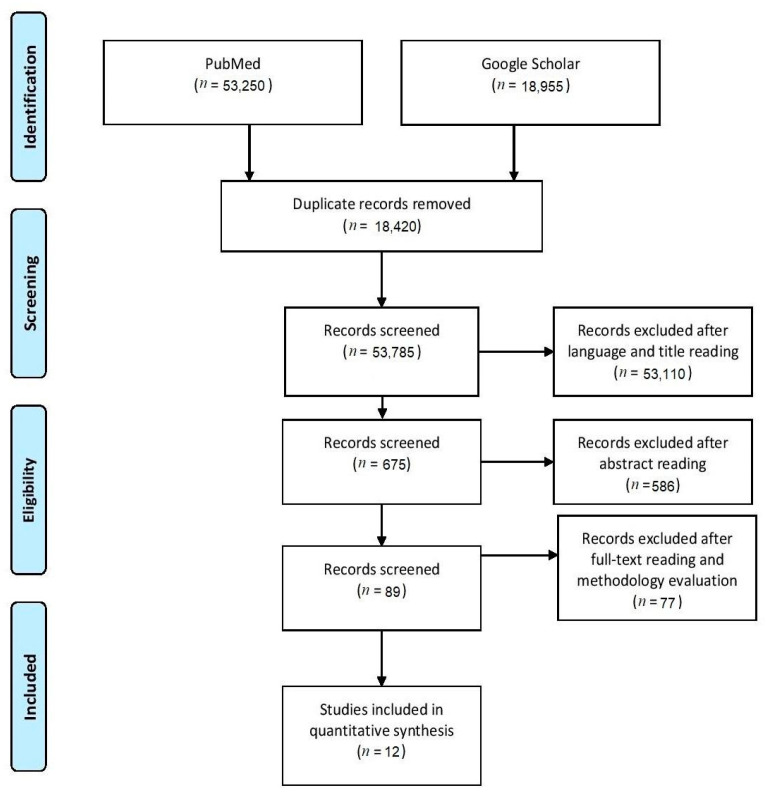
PRISMA flow diagram of the literature search and the studies selection process.

**Figure 2 ijerph-20-00453-f002:**
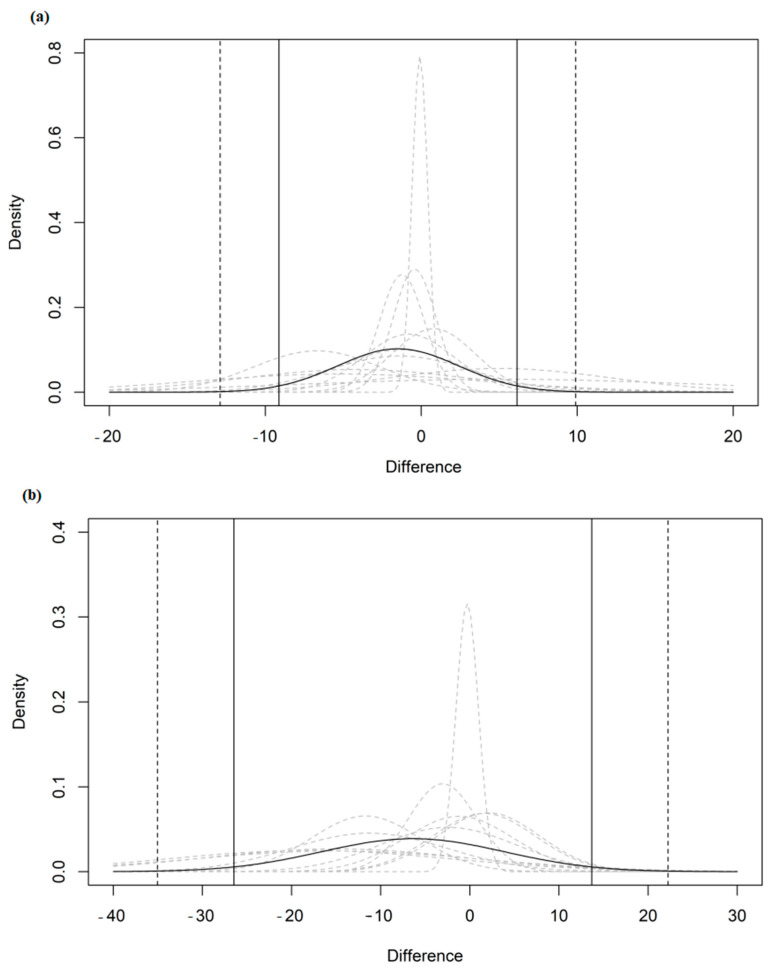
Pooled (black curve) and individual studies’ (grey dashed curve) estimated normal distributions of the differences between AerT and FAT_max_ expressed as mL/kg/min (**a** panel) and %VO_2max_ (**b** panel). NOTE: Vertical lines indicate the estimated pooled limits of agreement (solid vertical lines) and their outer CI 95% (dashed vertical lines).

**Figure 3 ijerph-20-00453-f003:**
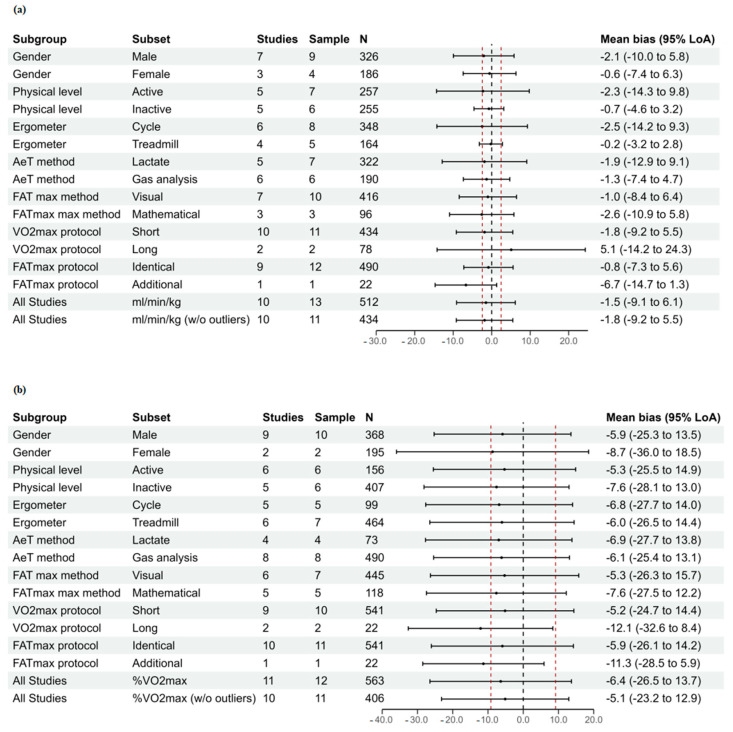
Forest plots of the pooled 95% limits of agreement (LoA) and mean bias with the estimated FAT_max_ zone (red dotted line a priori calculated to be ± 4.89/2 (± 2.44) mL/kg/min and ± 18.34/2 (± 9.17) %VO_2max_, see Appendix A) for exercise intensities expressed as mL/kg/min (**a** panel) and as %VO_2max_ (**b** panel) further divided into subgroups. NOTE: Sample: number of samples for each study. All study samples represent estimates including all studies; w/o outlier represents estimate performed excluding potential outliers.

**Table 1 ijerph-20-00453-t001:** Descriptive statistics of all included studies after the systematic review with identified categorical moderator variables, categorized by differences in study populations (clinical diversity) and characteristics of the studies (methodological diversity).

Trial	Trial ID	Group within Trial	N	Measurement Unit	Sex	Physical Level	Ergometer	AeT Detection Method	VO_2max_ Protocol	FAT_max_ Detection Method	FAT_max_ Protocol
1	Achten et al., (2004) [41]	a	33	b/min	Male	Active	Cycle	Lactate	Short	Visual	Identical
		b	33	L/min	Male	Active	Cycle	Lactate	Short	Visual	Identical
2	Bircher et al., (2005) [50]	a	48	mL/min/kg	Male	Active	Cycle	Lactate	Short	Visual	Identical
		b	48	b/min	Male	Active	Cycle	Lactate	Short	Visual	Identical
		c	48	b/min	Male	Active	Cycle	Lactate	Long	Visual	Identical
		d *	48	mL/min/kg	Male	Active	Cycle	Lactate	Long	Visual	Identical
		e	30	mL/min/kg	Female	Active	Cycle	Lactate	Short	Visual	Identical
		f	30	b/min	Female	Active	Cycle	Lactate	Short	Visual	Identical
		g	30	b/min	Female	Active	Cycle	Lactate	Long	Visual	Identical
		h *	30	mL/min/kg	Female	Active	Cycle	Lactate	Long	Visual	Identical
3	Emerenziani et al., (2019) [25]	a	52	b/min	Female	Inactive	Treadmill	Gas analysis	Short	Mathematical	Identical
		b	52	mL/min/kg	Female	Inactive	Treadmill	Gas analysis	Short	Mathematical	Identical
		c	52	%HR_max_	Female	Inactive	Treadmill	Gas analysis	Short	Mathematical	Identical
		d	52	%VO_2max_	Female	Inactive	Treadmill	Gas analysis	Short	Mathematical	Identical
4	Gonzalez-Haro (2011) [29]	a	11	%VO_2max_	Male	Active	Cycle	Lactate	Long	Mathematical	Identical
		b	11	%VO_2max_	Male	Active	Cycle	Lactate	Long	Mathematical	Identical
5	Michallet et al., (2008) [26]	a	14	L/min	-	Active	Cycle	Gas analysis	Short	Mathematical	Additional
		b	14	L/min	-	Active	Cycle	Lactate	Short	Mathematical	Additional
6	Nikolovski et al., (2021) [16]	a	22	mL/min/kg	Male	Active	Cycle	Gas analysis	Short	Mathematical	Identical
		b	22	%VO_2max_	Male	Active	Cycle	Gas analysis	Short	Mathematical	Identical
		c	22	%HR_max_	Male	Active	Cycle	Gas analysis	Short	Mathematical	Identical
7	Peric et al., (2018) [51]	a	57	mL/min/kg	Male	Active	Treadmill	Gas analysis	Short	Visual	Identical
		b	57	%VO_2max_	Male	Active	Treadmill	Gas analysis	Short	Visual	Identical
8	Peric et al., (2020) [52]	a	19	mL/min/kg	Male	Inactive	Treadmill	Gas analysis	Short	Visual	Identical
		b	19	%VO_2max_	Male	Inactive	Treadmill	Gas analysis	Short	Visual	Identical
9	Rynders et al., (2011) [28]	a	74	mL/min/kg	Male	Inactive	Cycle	Lactate	Short	Visual	Identical
		b	74	mL/min/kg	Female	Inactive	Cycle	Lactate	Short	Visual	Identical
10	Venables et al., (2004) [53]	a *	157	%VO_2max_	Male	Inactive	Treadmill	Gas analysis	Short	Visual	Identical
		b	143	%VO_2max_	Female	Inactive	Treadmill	Gas analysis	Short	Visual	Identical
11	Zurbuchen et al., (2020) [54]	a	22	mL/min/kg	Male	Active	Cycle	Gas analysis	Short	Mathematical	Additional
		b	22	%VO_2max_	Male	Active	Cycle	Gas analysis	Short	Mathematical	Additional
12	Chavez-Guevara et al., (2022) [55]	a	18	mL/min/kg	Male	Inactive	Treadmill	Gas analysis	Short	Visual	Identical
		b	18	%VO_2max_	Male	Inactive	Treadmill	Gas analysis	Short	Visual	Identical
		c	18	mL/min/kg	Male	Inactive	Treadmill	Lactate	Short	Visual	Identical
		d	18	%VO_2max_	Male	Inactive	Treadmill	Lactate	Short	Visual	Identical

NOTE: * effect sizes that resulted as potential outliers and excluded in the sensitivity analysis.

## Data Availability

Not applicable.

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
