# Peer review of "Factors Determining the Agreement between Aerobic Threshold and Point of Maximal Fat Oxidation: Follow-Up on a Systematic Review and Meta-Analysis on Association"

_ijerph, 2022, doi:10.3390/ijerph20010453_

Round 1

Reviewer 1 Report

The authors present a well written, robust attempt to exam the relationship between fatmax and ventilatory/aerobic threshold. While the background/introduction provides a robust examination and therefore justification of the available literature, there is a large gap within the proposed mechanisms/limitations and the testing methodology influence on indirect calorimetry. While I applaud the authors for their robust attempt to exam the relationship between the two variables. However, I encourage them to consider the variation within the testing methods. I have included some of the relevant literature that can either help reinforce their efforts or refute them. In either case, it is important the authors consider the cited literature below that will challenge the current status of their manuscript. I encourage the authors to reflect upon the current manuscript as the introduction/title of the paper is not relevant to the data/conclusions. Therefore, the current manuscript I do not believe is publishable. I do however, encourage the authors to exam what their central thesis is and review accordingly as much of the manuscript has great potential.

Additionally, there are minor grammatical error throughout.

METHODS

A robust methodological process is presented with great care to protect methodological validity. The authors acknowledge and present variation within included studies which is missing from the introduction.

RESULTS

Appropriate analysis of the data included.

DISCUSSION

The discussion resides around the methodological differences amongst the included papers, which is a reasonable/objective conclusion. However, the introduction/background literature is completely lacking this concept. The authors present literature background giving the impression of investigating AerT and fatmax. However, the discussion almost entirely resides around the lacking of homogeneity amongst the included papers and more specifically attempts to dissect differences such as training status, exercise mode, lactate measurement, etc.

Poole, DC, Jones, AM. Oxygen uptake kinetics. Compr Physiol 2: 933–996, 2012.

Purdom, T, Kravitz, L, Dokladny, K, Mermier, C. Understanding the factors that effect maximal fat oxidation. J Int Soc Sports Nutr 15: 3– 10, 2018.

Randell, RK, Rollo, I, Roberts, TJ, et al. Maximal fat oxidation rates in an athletic population. Med Sci Sports Exerc 49: 133–140, 2017

Robergs, RA, Dwyer, D, Astorino, T. Recommendations for improved data processing from expired gas analysis indirect calorimetry. Sports Med 40: 95–111, 2010. 26.

Author Response

Dear Reviewer,

Thank you for your suggestions and the time taken to read this work. It is highly appreciated.  Our paper is the second part of a two-piece paper investigating the association and agreement between FATmax and AerT. We aimed at investigating the high variability in the previously published results from a statistical perspective. We did such by identifying the moderators and their subgroups, and a third variable that affects the strength of the relationship between FATmax (dependent variable) and AerT (independent variable) (Cohen, Jacob; Cohen, Patricia; Leona S. Aiken; West, Stephen H. (2003)), applied multiple regression/correlation analysis for the behavioral sciences (Hillsdale, N.J: L. Erlbaum Associates) since moderators could be related to either clinical or methodological diversity between the studies (Higgins, J.P.T.; Thomas, J.; Chandler, J.; Cumpston, M.; Li, T.; Page, M.J.; Welch, V.A. Cochrane Handbook for Systematic Reviews of Interventions. 2nd Edition. Chichester. 2019).

Introduction:

Reviewer #1 recommended to consider variations within the testing methods and commented “ ... there is a large gap within the proposed mechanisms/limitations and the testing methodology influence on indirect calorimetry.”.  In response to those comments, we rewrote parts of the introduction and discussion sections to better clarify our study aims as we assessed methodological differences used to determine FATmax and AerT between the studies.

We thank the reviewer for proposing literature that would improve the quality of our manuscript (Poole, Purdom, Randell, and Robergs). However, we believe that they are not relevant to our work as they only address FATmax (Purdom and Randel) and don't consider FATmax nor AerT (Poole is about exercise prescription/physiology and Robergs is about filtering techniques of indirect calorimetry data) from the perspective of the agreement between FATmax and AerT. For some of the suggested literature, we can only presume that this reviewer considered it an importance that different raw data editing techniques could affect FATmax or AerT location and therefore should be taken into consideration. However, such information was not reported in included studies, and hence, we were unable to analyze it. Concerning methods used to identify FATmax, this specific topic is addressed as a moderator and included in our analysis. Nevertheless, some of the suggested literature is now added to the paper. Noteworthy, our particular workgroup has previously examined the influences of differences in indirect calorimetry systems and stoichiometric equations on the FATmax (Peric et al., A systematic comparison of commonly used stoichiometric equations to estimate fat oxidation during exercise in athletes. J Sports Med Phys Fitness. 2021;61(10):1354-1361) which we consider might be of interest for the Reviewer.

We only addressed the variability between the included studies in the discussion section (not in the introductions) as we became aware of the articles causing such variability only after completing the Systematic Review and obtaining the results.

Discussions:

We believe that the edits we made in our introduction and discussion sections addressed the reviewer's comments.  We clarified that we aimed at examining the methodological differences between studies that influenced the agreement between the FATmax and AerT. We did so by investigating the influence of different methodological approaches on the agreement and not by independently assessing FATmax or AerT. Our introduction describes the inter-subjects variability and challenges related to the identification of FATmax, the importance of training at FATmax, the benefits of threshold training, and explains that methodological differences (moderators) between studies can influence the association between FATmax and the AerT. We hypothesized that different moderators can and will affect the agreement as previously showed to be in the case of the association between AerT and FAT. In our discussion, we elaborated on the influence of different moderators on the agreement following our stated hypothesis.

We genuinely hope these explanations as well as the changes made directly in the paper have provided sufficient explanations and clarifications on this main paper objective and cleared any doubts.

Reviewer 2 Report

1. The introduction of the included population of related articles is not detailed enough. The range of fat oxidation rate and maximum fat oxidation intensity of different types of population is different. The article distinguishes between gender and physical activity level. Does it include athletes (sprinter, endurance runners, or other types of athletes)?

2. The test protocol included in the study stated only the type of motion used and what the specific steps of the test were.Need to add some content,e.g., at what speed the test was started, how long each speed lasted, what equipment was used to collect the gas metabolism, etc. 

In my opinion, the above two points will affect the results of this study, so I suggested to elaborate in the table or appendix of this paper.

Author Response

Dear Reviewer,

Thank you for your comments and suggestions, they are appreciated. The authors considered making such considerations but they had to be left out due to a purely statistical problem. Even though we were unable to analyze these particular data, we have added them to the paper.

  1. We provided additional considerations concerning the population in the discussion section (after completing the systematic review) as we couldn’t make them in the introduction section due to this being a SR.

We fully acknowledge the Reviewers statement regarding fat oxidation rate variability and this is something we have additionally addressed in both the introduction and discussion sections (in red). However, please consider that fat oxidation rates (MFO) are not directly related to the topic of the paper.

As for requests to further distinguish between active population (e.g., athletes type) and their sport type origin (i.e., sprinters, endurance, team sports), this approach is simply not possible due to statistical issues. For us to perform a multilevel meta-analysis and to examine an agreement between FATmax and AerT, we need to identify moderators followed by subgrouping, a third variable that affects the strength of the relationship between a dependent variable (FATmax) and an independent variable (AerT) (Cohen, Jacob; Cohen, Patricia; Leona Aiken; West, Stephen H. (2003). Applied multiple regression/correlation analysis for the behavioral sciences. Hillsdale, N.J: L. Erlbaum Associates). Moderators could be related to either clinical or methodological diversity between the studies (Higgins, J.P.T.; Thomas, J.; Chandler, J.; Cumpston, M.; Li, T.; Page, M.J.; Welch, V.A. (editors). Cochrane Handbook for Systematic Reviews of Interventions. 2nd Edition. Chichester. 2019). Clinical diversity between the included studies is related to the subjects, their gender, and physical activity levels. Subgroups should be at least two, however, for statistical findings to be robust it is recommended to have at least 3 effect sizes (samples) within the subgroup (Douglas, B.; Mächler, M.; Bolker, B.; Walker, S. Fitting Linear Mixed-Effects Models Using lme4. J. Stat. Softw.2015,67, 1–48). This is why, when possible, we created subgroups having more effect sizes within the group, and hence had all active participants grouped in one subgroup and inactive in another. There was just too large difference within included population not allowing us to analyze them separately. However, to somehow address suggestions made by the Reviewer, we have informed future readers regarding diversity within the active participants in the results and discussion sections even though these data were unable to be analyzed.

The same issue is related to the testing protocol and our decision to group them into short (< 3min) and long stages (>3min). In this study we considered trials using 1- to 5- min stages and no two studies used identical protocols. Even when studies using the same stage length were found, the authors used different ergometers, starting speed or power, or speed or power increments at every stage. If we were to analyze these differences, we would have had several moderators and several subgroups all with only one study which eventually would have limited the robustness of the results, making them statistically insignificant and most important, incorrect.  We have addressed this issue in our limitation section. Nevertheless, to account for the Reviewer's suggestions, we have presented this variability in our results.

We hope these answers have justified your concerns.

Reviewer 3 Report

The main question addressed by the research: If the aerobic threshold and the point of maximum oxidation of fats present agreement (LoAs), since other studies had only evaluated the correlation between both phenomena.

The topic is original, using statistical methods different from previous studies, the authors analyzed the agreement between the aerobic threshold and the point of maximum fat oxidation. This analysis was performed through a systematic review study.

Compared with other published material, the authors used different statistical techniques (they calculate the Bland-Altman meta-analyses LoAs). This procedure highlights the work.

 No further improvements should be considered regarding the methodology, the methods are adequate.

 There is a relationship between the objectives and conclusions of the study. They are adequate.

 The references are appropriate. The figures are adequate.

General questions

Congratulations for the excellent work

 Minor questions

-  Reference: please, review all references: a) some titles were written with the first letter in capital letters and others in lower case (need to standardize);

Author Response

Dear Reviewer,

Thank you for your acknowledgment of our work, it is highly appreciated.

We have corrected the references as required.